# The percentile-matching technique for synthetic eye tracking signal degradation: A biometric case study

Henry Griffith, Dmytro Katrychuk[ID]☯*, Mehedi H. Raju[ID]☯, Samantha Aziz‡, Dillon J. Lohr‡, Oleg V. Komogortsev

Department of Computer Science, Texas State University, San Marcos, Texas, United States of America

☯ These authors contributed equally to this work.
‡ These authors also contributed equally to this work.
* d_k139@txstate.edu

## Abstract

This manuscript demonstrates an improved model-based approach for synthetic degradation of pre-recorded eye movement signals. Recordings from a high-quality eye tracking sensor are transformed to make their eye tracking signal quality resemble ones captured on a lower-quality target device. The proposed model improves the realism of the degraded signals versus prior approaches by introducing a mechanism for degrading spatial accuracy and temporal precision. Specifically, a percentile-matching technique is developed for mimicking the relative distributional structure of the target data signal quality characteristics. The model is demonstrated to improve realism on a per-feature and per-recording basis using data from EyeLink 1000 and SMI eye tracker embedded within a virtual reality platform. This study is first to show that the percentile-matching technique enables more accurate approximation of the target set using the biometric user authentication as an end task. Its mean absolute error of estimating the target set performance level is better for all three metrics considered (compared to the baseline) - a) 0.26 (versus 0.84) of d-prime; b) 12.08% (versus 31.10%) of False Rejection Rate at 1-in-10000 False Acceptance Rate; c) 3.08% (versus 4.69%) of Equal Error Rate. This paper also expands eye movement literature by suggesting an application-agnostic realism assessment workflow based on the 1 Nearest-Neighbor classifier. Our model improves the related median classification accuracy metric by 35.7% versus the benchmark model towards the ideal 50% value.

## Introduction

Eye tracking (ET) technology has been in use in laboratory settings for various value-added applications for decades, including health assessment [1], education [2], and marketing [3]. More recently, ET sensors are being deployed as an input modality in consumer electronic devices within various extended reality domains [4–7]. As the

**Data availability statement:** GazeBase: https://doi.org/10.6084/m9.figshare.12912257.v3 GazeBaseVR: https://doi.org/10.6084/m9.figshare.21308391.v2.

**Funding:** Authors Samantha Aziz and Dillon Lohr are recipients of National Science Foundation Graduate Research Fellowship under Grant No. DGE-1840989 and DGE-1144466, respectively. Any opinion, findings, and conclusions or recommendations expressed in this material are those of the authors(s). The funders had no role in study design, data collection and analysis, decision to publish, or preparation of the manuscript.

form-factor of the target integration platform changes, the corresponding signal quality (e.g., spatial accuracy and precision, temporal precision, etc. [8]) provided by the ET sensor also typically varies [9,10], thereby limiting the generalization of signal processing workflows across hardware platforms. Moreover, the development of novel ET sensors is complicated by high cost associated with the initial ET data collection, especially on the scale large enough to support modern machine learning based application pipelines.

To address the design challenges associated with data scarcity and variability in signal quality, various model-based techniques for transforming ET signals have been previously proposed [11,12]. These algorithms allow for data recorded using a particular (source) ET device to appear as if they were instead recorded on a different (target) device [13]. These transformations are valuable for synthetically augmenting the scale and diversity of available training data for machine learning applications [11]. In addition, they allow system designers to estimate performance variability for various target applications as a function of ET sensor quality [14]. Such transformations may either explicitly utilize the performance characteristics of the target hardware as parameters (e.g., the simplistic downsampling of data to a target rate), or develop estimates of the underlying hardware style by modeling a representative target data set. An alternative approach that has been gaining traction in the literature is to synthetically expand the target set via signal generation techniques, with the most successful recent solutions utilizing machine learning [15–17]. However, we focus on the source-to-target transformation methods with the ability to directly and systematically control the quality metrics of transformed ET signals, as we are not aware of any existing end-to-end learning solution capable of that yet.

The synthetic transformation of ET signals is complicated by several factors. Namely, the metrics and corresponding definitions used to assess device quality vary considerably across the research community [14,18]. In addition, the signal deviation from the expected value may be associated with both noise from the recording instrument and the human visual system (HVS) itself [19,20]. To address this latter concern, recent models have suggested injecting noise with a uniform power spectral density, which has been shown to resemble the instrument noise component through the assessment of artificial eye recordings [18,21]. While this strategy improves the realism of the relative instrument-to-HVS noise levels under the stated assumptions, it does not support the modulation of both the spatial accuracy and temporal precision of the source data. Moreover, it lacks the ability to replicate the distributional structure of the desired performance metrics across a target data set.

The research described herein demonstrates a series of improvements for a previously proposed additive white noise degradation model [11] which enhances the realism of the synthetically generated signals. Namely, modifications are introduced to support the marginal degradation of both the spatial accuracy and temporal precision of the source data set. Moreover, a percentile-matching technique is proposed in order to mimic the underlying distributional structure of the target data set quality metrics across recordings. The proposed transformation workflow is validated using subsets of GazeBase [22] as the source data set and GazeBaseVR [23] as the target data set. The realism of the transformation is assessed by comparing quality

metrics computed on the synthetically generated data to those computed on the target data on both a per-feature and per-recording basis.

A big potential utility of the proposed degradation pipeline lies in approximating an end task performance given only the distributional characteristics of the target data signal quality metric (e.g., spatial precision). Its corresponding percentile profile could be synthetically made to represent a hypothetical target device or it could be estimated for a real target device from a limited set of data samples. The raw source data set is then synthetically degraded to resemble the data quality of the target one. An end task could be subsequently performed using the latter. The closer the resulting evaluation metrics end up being to those obtained on the target set - the better chosen synthetic degradation method is.

We selected the biometric user authentication as a specific end task to benchmark the degradation methods considered in this study. The eye movements were established as a viable biometric source in the seminal work of [24]. Its main advantage is spoof-resistance, as the gaze patterns stem from a complex combination of behavioral and physical human traits, the model for which is hypothesized to be infeasible to forge. The ET sensors continue to emerge in modern extended reality hardware, such as Meta Quest Pro released in 2022 [25], Sony PSVR2 (2023, [26]), and Apple Vision Pro (2024, [27]). The wearable form of these devices with the gaze being a natural input modality makes the eye movement biometrics even more promising, especially as a solution for continuous authentication [28].

The primary contributions of this work include - 1) a modified mechanistic transform model which offers enhanced parameterization and realism for the model-based synthetic degradation of eye tracking signals; 2) a case study in eye movement biometrics evaluating the performance on synthetically degraded data in an end task of biometric user authentication. To our best knowledge, this is the first study to show an improved approximation of the application-specific performance level of lower-quality real data, compared to a simplistic white noise degradation prevalent in the ET literature; and 3) an application-agnostic assessment workflow suitable for assessing the realism of synthetically generated signals on a per eye tracking signal quality metric and per-recording level using a 1 nearest-neighbor (1-NN) classifier. While the efficacy of this technique has been previously demonstrated for images generated using generative adversarial networks, we are unaware of its application for the assessment of synthetically generated eye movement signals.

## Methods

### Data

A subset of the publicly-available GazeBase repository [22] was utilized as the source data within this project. GazeBase is a multi-stimuli, large-scale, longitudinal data set of monocular (left) eye movement recordings captured over a 37-month period from 322 college-aged subjects. Each collection period (denoted as a round) consists of a pair of contiguous recording sessions, during which subjects completed a battery of common tasks across both sessions. Tasks included fixation, random saccade, video viewing, reading, and interactive game play. A subset of subjects from each prior round completed subsequent rounds of recording, with a total of nine rounds of recording performed.

Eye movements were captured in GazeBase using the EyeLink 1000 eye tracker at 1,000 Hz [29]. For the current analysis, only random saccade task recordings from both sessions of Round 1 were included for all 322 subjects. This one task was chosen as it was the most complex guided-viewing task with specified stimulus locations, which are necessary due to the definitions of the employed ET signal quality metrics described in [14].

The target data employed herein was captured using a subset of the publicly-available GazeBaseVR repository [23]. The experimental battery for this collection was chosen to align with the GazeBase collection procedure as much as possible (e.g., utilization of similar tasking, contiguous recording sessions within individual rounds, etc.). Similar to GazeBase, GazeBaseVR is a multi-stimuli, longitudinal data set of binocular eye movement recordings captured over a 26-month period from 407 college-aged subjects. Each round comprises a pair of contiguous recording sessions where subjects completed a battery of five tasks. Tasks for GazeBaseVR included vergence, smooth pursuit, video viewing, reading, and random saccade. Each subject completed up to a total of 3 rounds.

Eye movements in GazeBaseVR were captured using a modified HTC Vive with an embedded SMI eye-tracking device at 250 Hz [23]. Recordings from a random subset of 322 subjects (to match the number utilized in the source GazeBase set explained above) captured while completing the random saccade task during both sessions of Round 1 were utilized herein. It should be explicitly noted that both the subject pool and target (stimulus) trajectories of the random saccade task in GazeBaseVR are disjoint from those utilized in GazeBase. Namely, targets remained stationary during the random saccade task for a fixed period of time (1000 ms) for GazeBase and for a random period of time (uniformly distributed from 1000–1500 ms) for GazeBaseVR. In addition, the target pattern was varied across the two experiments, and no common subjects were utilized across the two collections. As the technique described herein compares signals in terms of data quality metrics which are computed upon fixations, the aforementioned differences in stimuli trajectories should not significantly influence the resulting analysis.

Although GazeBaseVR captures binocular gaze data, only data from the left eye was utilized within this analysis. Monocular data is reported by the modified HTC Vive at a nominal sampling rate of 250 Hz, with some variation around the ideal intersampling interval (ISI) of 4 ms observed in the resulting data. This variability is in contrast to the data reported by the EyeLink 1000 device, which has a constant ISI of 1 ms. Additional information regarding the achievable quality levels of the HTC Vive platform may be found in [14] and [23].

**Eye tracking signal quality metrics**

Both per-channel and combined spatial accuracy and precision, along with temporal precision, were used to characterize ET signal quality within this workflow. These metrics were chosen as they represented the best inherent summary characterization of the eye tracking sensor. Each metric was computed using the definitions described in [14]. Namely, spatial accuracy is defined as the systematic bias of the gaze measurements produced by an eye tracker about the ground truth value, while precision describes the corresponding dispersion of these measurements about their central tendency. Temporal precision describes the variation in the ISI about the nominal value of the device.

The computation of ET signal quality metrics requires the estimation of temporal domain of each fixation. In the absence of algorithmic classification, as suggested in [14], this was achieved by estimating the average saccade latency on a per-file basis in order to parse the fixation intervals on each stimulus. Saccade latency accounts for the delay between stimulus transitions and the corresponding saccade initiation, which is typically on the order of 200 ms [30]. Saccade latency was estimated by finding the shifted value of the gaze signal which exhibited minimum separation from the target signal in the Euclidean sense (see Fig 5 of [14] for a visual summary of this process).

The start and stop times for each fixation were estimated based upon adjusting the target transition times according to the estimated saccade latency on a per-recording basis. Next, the first 400 ms after the fixation start were discarded to account for inter-recording variability in saccade latency. Namely, the removal of this portion of the signal helps to ensure that only valid fixation intervals are captured for subsequent computation of the data quality metrics. Once this initial offset was discarded, the following 500 ms were selected for further processing.

Within each candidate fixation, samples with a distance-to-centroid either outside Tukey's fences [31] or greater than 2 degrees of the visual angle (dva) were marked as outliers. Once outliers had been removed, the spatial accuracy of each fixation was computed on a per-channel and combined basis according to Eqs 1–2.

$$\theta_h = \frac{1}{n}\sum_{i=1}^{n}|x_i^g - x_i^t|; \ \theta_v = \frac{1}{n}\sum_{i=1}^{n}|y_i^g - y_i^t| \tag{1}$$

$$\theta_c = \frac{1}{n}\sum_{i=1}^{n}\sqrt{(x_i^g - x_i^t)^2 + (y_i^g - y_i^t)^2} \tag{2}$$

where $\theta_h$, $\theta_v$, and $\theta_c$ correspond to the horizontal, vertical, and combined spatial accuracy in dva, respectively; $x^g$ and $x^t$ correspond to the gaze and target samples in the horizontal channel, respectively; $y^g$ and $y^t$ are analogous to $x^g$ and $x^t$ for the vertical channel; and $n$ corresponds to the number of samples within the fixation. In a similar fashion, the spatial precision of each fixation was computed as shown in Eqs 3–4.

$$MAD_h = M(|x_i^g - M(x^g)|); \; MAD_v = M(|y_i^g - M(y^g)|) \tag{3}$$

$$MAD_c = \sqrt{M(|x_i^g - M(x^g)|)^2 + M(|y_i^g - M(y^g)|)^2} \tag{4}$$

where $M()$ denotes the median operator, and $MAD_h$, $MAD_v$, and $MAD_c$ denote the horizontal, vertical, and combined spatial precision, respectively.

Temporal precision was computed by taking the standard deviation of the sample-over-sample difference in timestamps on a per-file basis.

## Benchmark transformation model

The mechanistic data degradation model described in [11] was implemented herein for initial benchmarking purposes. While it was attempted to replicate the implementation as closely as possible to the procedure described in the original manuscript, some modifications were made to enhance performance for the current application as described in the remainder of this section. This modified model is hereby denoted as the benchmark model.

The benchmark model implements independent mechanisms to reduce bandwidth and degrade the spatial precision of the source data set. Bandwidth is reduced through a custom downsampling procedure intended to avoid aliasing in the resulting signal. The initial implementation of the benchmark model filtered the source signal using a Butterworth filter with a cutoff rate set at 0.8 times the Nyquist frequency associated with the target sampling rate. For the current study, this filter was modified to a zero-phase architecture to minimize the delay between the filtered and original signal. After filtering, the signal is resampled at the target sampling rate using first-order spline interpolation.

Spatial precision is degraded in the benchmark model using an additive Gaussian noise stationary process as shown in Eqs 5–6. As noted in Eq 6, the variance of the additive noise may be tuned to increase towards the screen edges using a Gaussian weighting function.

$$x_t = x_s + N_x; \; y_t = y_s + N_y; \; N_x, N_y \sim \mathcal{N}(0, \sigma^2(x_s, y_s)) \tag{5}$$

$$\sigma^2(x_s, y_s) = \alpha(x_s, y_s) \times \sigma_0^2; \; \alpha(x_s, y_s) = \exp\left(\frac{-(r_s - r_{max})^2}{2\sigma_s^2}\right) \tag{6}$$

where $x_t$ and $x_s$ correspond to the synthetic target and real source samples in the horizontal channel, respectively; $y_t$ and $y_s$ are analogous to $x_t$ and $x_s$ for the vertical channel; $\sigma_0$ denotes the maximum value of the additive noise variance; $\sigma_s$ denotes the dispersion parameter in the Gaussian weighting function; $r_{max}$ denotes the maximum radial dimension of the screen in dva; and $r_s = \sqrt{x_s^2 + y_s^2}$. In this work, the scaling function $\alpha$ was removed, as the preliminary analysis of GazeBaseVR subset used as target data demonstrated no variability in precision as a function of eccentricity.

## Benchmark model tuning

Tuning the benchmark model in its native implementation requires specification of the nominal sampling rate of the target eye tracker (250 Hz), along with determination of the additive noise variance parameter $\sigma_0^2$.

For the current analysis, this latter value was tuned by evaluating the resulting spatial precision of a sample of transformed source signals produced using varying values of this parameter. Moreover, upon visualizing the resulting marginal combined precision degradation as a function of $\sigma_0^2$, a linear relationship was observed. Based on that finding, a linear regression was utilized to develop an estimate of this parameter:

$$\sigma_0^2 = 1.2904 * \Delta MAD_{c_p} + 0.0356 \tag{7}$$

where $\Delta MAD_{c_p}$ denotes the required marginal degradation computed as a difference between the target and the source combined spatial precision $MAD_c$ value, each computed at the percentile rank $p$ within corresponding set.

## Development of modified degradation model

The baseline model is characterized by several notable limitations. First, as the additive noise parameter utilized is applied constantly across each source recording, the corresponding distribution of the spatial precision of generated target data only matches that of the source data in the central tendency set (e.g., the relative dispersion across files is distorted). This phenomenon is demonstrated in Fig 1 for the case of combined spatial precision, where the resulting synthetic data retains the distribution of the source data with its central tendency shifted to match the target data. Second, as the baseline algorithm does not marginally degrade either the spatial accuracy or temporal precision, the resulting synthetic data is not capable of exhibiting resemblance to the target with respect to these measures.

To address these limitations of the baseline model, various modifications were implemented. To retain the distributional structure of the spatial precision of the target data set, a percentile-matching technique was implemented as shown

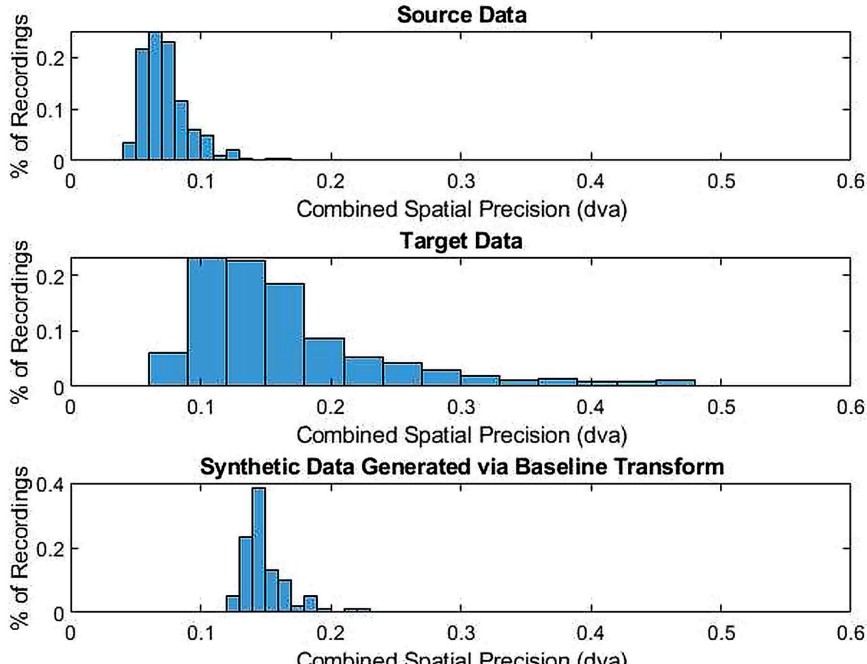

**Fig 1**. **Distribution of combined spatial precision across files for the target (top) and synthetically degraded data (bottom).** The benchmark model matches the target spatial precision only in central tendency, while the original source distribution shape is maintained.

in Fig 2. In this technique, the percentile ranking of the combined spatial precision within the source distribution for each file is initially computed. The spatial precision of the matching percentile within the target distribution was then determined in order to compute the requisite marginal precision degradation, which is subsequently used to compute the requisite additive noise parameter using the linear regression model discussed in Sect Benchmark model tuning.

In short, the original benchmark model (Fig 1) fails to achieve the proper distribution of a target data quality metric, a deficiency addressed by the proposed percentile-matching method (Fig 2; shown for the case of combined spatial precision).

A similar percentile-matching approach was also utilized to marginally degrade the spatial accuracy of the source signals to resemble the corresponding distribution within the target data set. Namely, the marginal accuracy degradation for a given source file was initially determined on a per-channel basis. Then, for each fixation within the source file, a specific per-channel degradation value was drawn from a normal distribution centered around the value computed in the previous step. The standard deviation of this distribution was chosen such that 99.7% of the values were within 20% of the requisite accuracy degradation value. The sampled value was subsequently added to the gaze signal during the fixation after weighting by a uniformly distributed random sign (e.g., $+/-$) value. This randomness (e.g., both variability from the requisite marginal degradation value and in offset direction) was used as a naïve initial approach to improve realism of the resulting synthetic signals. To determine fixation boundaries for adding the marginal accuracy degradation signal, a shift-based alignment approach similar to that described above and in [14] was employed (e.g., the marginal accuracy degradation signal was added to the latency-adjusted gaze signal which was defined as the shifted gaze signal which minimized the Euclidean distance computed between the target and gaze value).

To replicate the degradation in temporal precision observed in the target data set, the domain values for the re-sampling procedure in the original model were perturbed by adding a zero-mean random Gaussian noise term to each nominal time stamp. The standard deviation of this distribution was set equal to the empirical value observed variation in the ISIs from the nominal sampling period (i.e., 4 ms) of the target set.

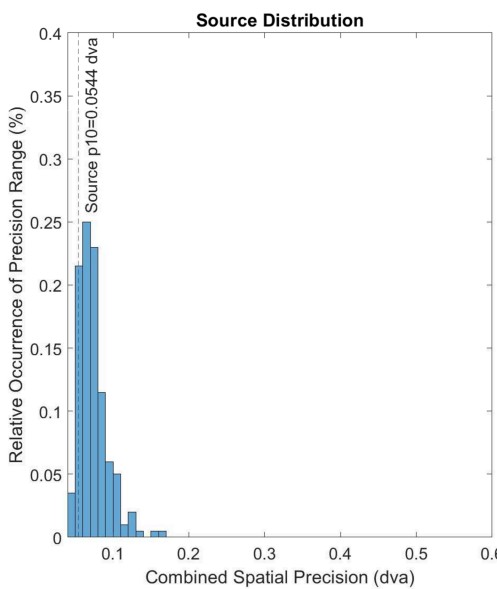 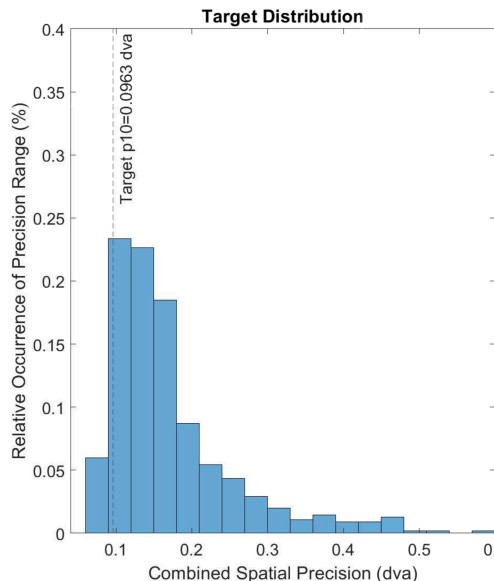

**Fig 2**. **The proposed technique matches the underlying distributional structure of a given metric across files.** For each source file, the relative percentile rank of the metric is initially computed. Based upon matching the same percentile rank in the target data set, the target metric value is determined, with the requisite tuning parameter $\sigma_0^2$ to achieve this marginal degradation estimated using Eq 7.

The various changes implemented within the modified Zemblys et al. [11] algorithm are summarized below.

- A percentile-matching technique was proposed to match the distributional structure of the spatial precision observed in the target data set.
- A similar technique for degrading the spatial accuracy was implemented, while the degradation of this data quality metric to mimic its distributional structure in the target set was not addressed in the original method.
- A technique for degrading the temporal precision also missing in the original method was implemented based on random Gaussian noise.

### Degradation realism assessment based on an end task performance

The earlier advances in eye movement biometrics relied on complex pipelines with a separate step of eye movement classification and a subsequent feature design based on the domain expert knowledge [32–34]. The more recent breakthroughs were achieved with deep end-to-end learning from raw position and velocity data with a minimal reliance on hand-crafted features [35] (with the follow-up in [36]), [37] (followed by [38]). We elected to use the framework of Eye Know Eye Too (EKYT) presented in [38], as it posted the state-of-the-art biometric performance on GazeBase, which was used in our study as a source set for data degradation. The EKYT model is based on the DenseNet deep convolutional architecture [39] and learns to transform gaze sequences into lower-dimensional embedding vectors using multi-similarity loss [40]. The method operates with a balanced set of negative sample pairs that come from different subjects and the positive pairs sampled from the same subject. During training, the optimization of the loss function results in pushing the embeddings from a negative pair further apart and reducing their cosine similarity, with the inverse being true for the positive pairs. The more in-depth review of the whole EKYT methodology can be found in [38].

We used the same subsets of GazeBase and GazeBaseVR in this biometric experiment as in the main degradation study. The GazeBase (i.e., source set) was degraded to match the data quality level of the GazeBaseVR (i.e., target set) using the baseline and the percentile-matching methods, producing two more respective synthetic data sets. Each of the total 4 sets was used independently, separate from each other, as explained further. The whole set of 322 subjects was split into the train and test sets using the 75%/25% ratio to enable the evaluation of 4 separate models in a cross-validation manner. The random saccades recordings from Round 1 Session 1 were used for training with the corresponding subject set. At test time, the Session 1 data of held out subjects were used for enrollment to create a database of user embeddings. The Session 2 data of the same subject set were then used to add user authentication embeddings to the database.

The performance of the resulting biometric system was evaluated using the test subject set as follows. The pairwise cosine similarities were computed between an enrollment embedding and an authentication embedding for each subject pair. The embeddings that came from different subjects create the distribution of impostor similarity scores, while those corresponding to the same subject form the genuine scores distribution. The first metric we report that could be already computed at this point is the d-prime (decidability index $d'$ as presented in [41]). The d-prime computes the degree of separation between the distributions of genuine and impostor scores based on their means and standard deviations. The higher d-prime value corresponds to a higher degree of separation, which should result in a better biometric system.

As the next step, the receiver operating characteristic (ROC) curve is constructed using the genuine and impostor scores as positive and negative samples, respectively. Each point on the ROC curve is a particular similarity threshold that could be used as a decision boundary for a successful authentication into a biometric system. It also represents a trade-off between the failed authentication attempts of genuine users (false rejection rate or FRR) and erroneous acceptance of impostors into the system (false acceptance rate or FAR). The remaining two metrics that we use for biometric performance reporting are the particular points on the ROC curve established in the biometric literature. First, the

point where the FRR and FAR are the same corresponds to Equal Error Rate and is a common single measure of biometric performance, with the value of 0% representing the ideal case when the genuine and impostor users are always correctly identified. Second, it is common to consider the FRR at a fixed FAR level chosen according to pre-defined security requirements or to reflect a performance level of a comparable system [42]. We follow [38] in reporting the FRR at a 1-in-10000 FAR, as it relates to the security level of a 4-digit PIN prevalent in day-to-day life.

One of important hyperparameters of a biometric system is the data length used for enrollment and authentication, which plays major role in forming the trade-off between system's accuracy and operation time. The employed biometric framework of EKYT was designed to operate on an input window of fixed length. However, it is possible to utilize more samples for model inference by splitting the available data into continuous chunks matching the length used during training. Then, the corresponding set of embeddings is collected by separately passing each chunk through the EKYT model. The single representative embedding is finally produced through aggregation. The same values for all hyperparameters were used as reported in the original study [38], including the input window length of 5 seconds.

### Application-agnostic realism assessment

The realism of the synthetically generated signals was also assessed on an aggregate basis using a 1-NN two-sample test as described in [43]. Namely, a 1-NN classifier was employed on ET signal quality feature vectors of the target and synthetic data sets using a leave-one-sample-out strategy. For an ideal transform scenario, the relevant classification accuracy would be 50%, corresponding to the inability of the classifier to distinguish between synthetic and real data at a performance level better than chance. As this method requires the size of the two sample sets to be identical, a subset of the target data was randomly sampled for comparison. To elucidate the effect of this sampling, the resulting analysis was repeated five times for random samples from both tested degradation models.

## Results

### Comparison of individual eye tracking signal quality metrics

The ET signal quality of data generated using the proposed and benchmark model was compared to the real target data set in order to assess the efficacy of the two approaches. A visualization of precision-related metrics across the two synthetic data sets and target data set is shown in Fig 3. A similar presentation of accuracy-related metrics is shown in Fig 4. As shown, the distributions in the synthetic data produced by the modified workflow exhibit significantly improved resemblance to the target data when compared to the baseline model. It should be noted that temporal precision metrics are not visualized since the baseline model has no mechanism for degrading this parameter (i.e., source data transformed using the baseline model would exhibit no variability in temporal precision, making visual comparison with the target data set trivial).

### Biometric performance on degraded data

The entire 60 seconds of the random saccades task recording from Session 2 could be used for evaluation, but using all data is rarely practical. Moreover, there is typically a point after which adding more data stops bringing significant performance improvements. That optimal input length varies by data set and should be tuned for each individual case. Rather than relying on the literature [38], we chose to investigate the variability in biometric performance as a function of input sequence length. As such, these numerous evaluation scenarios provided a more robust picture of how well each tested degradation method approximates the biometric performance on the target set.

To ensure optimal data utilization, the 60 seconds of recording were split into non-overlapping chunks of 5 seconds. All input sequence length settings ranging from 1 to 12 were tested. Note that in each case the resulting concatenation

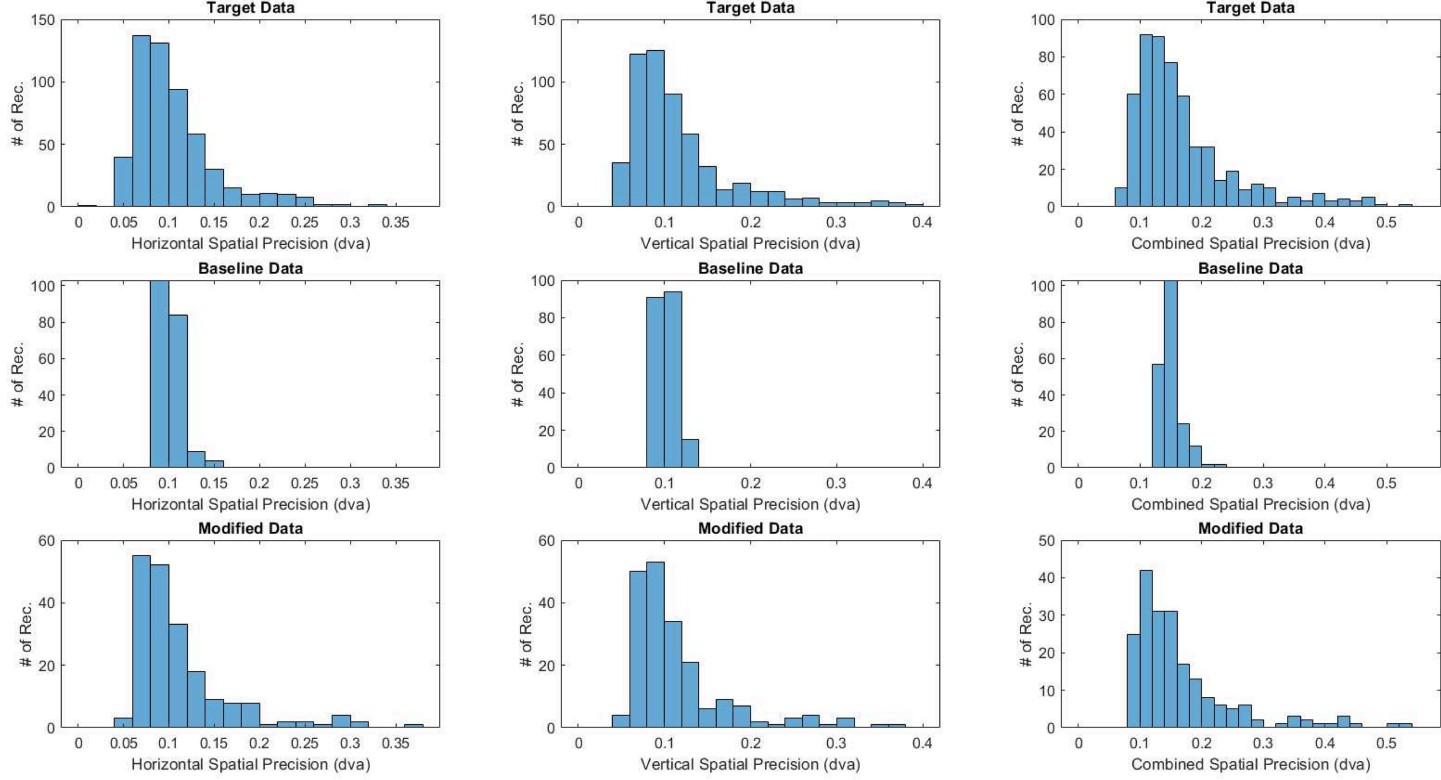

**Fig 3**. Variation in spatial precision across the target and synthetic data sets produced by the baseline and modified model, respectively.

of chunks began at the recording start and was continuous from then on (i.e., a newly added chunk always started at the rightmost end of the prior sequence). The aggregation of each resulting set of embeddings was computed using a per-feature mean.

The results of our biometric study on degraded data are shown in Figs 5, 6 and 7. The three figures depict the trends of each evaluation metric with varying input sequence length for all four tested data sets. Each evaluation scenario is summarized using a barplot with the mean and standard deviation computed over the four test folds. The results on data sets produced by two tested degradation methods are both treated as estimators of target set performance, which aligns with their intended use case. We begin our analysis with d-prime - the metric that summarizes the level of separation between genuine and impostor similarity score distributions (higher is better). The mean absolute error of estimating d-prime on the target set using data degraded with percentile-matching was 0.26 (averaged over every tested sequence length). This marks a substantial improvement over 0.84 mean error achieved with the baseline degradation method. In every tested scenario, our proposed degradation technique was closer to the d-prime achieved on the target set compared to the baseline. Interestingly, with the increase in input length, the baseline results converge with the source data set, while the proposed percentile-matching consistently closes the gap in d-prime with the target set.

Next, we move on to FRR (at a 1-in-10000 FAR) and EER metrics, which are more commonly used markers of biometric performance. Note that both are single-point measures of the ROC curve between FRR and FAR. Therefore, they are expected to be less robust to small changes in underlying genuine and impostor score distributions, compared to previously considered d-prime. The FRR has the most direct practical implication out of the three. It was chosen to show the percentage of erroneously rejected enrollments into the system when the rate of false acceptances is similar to a 4-digit

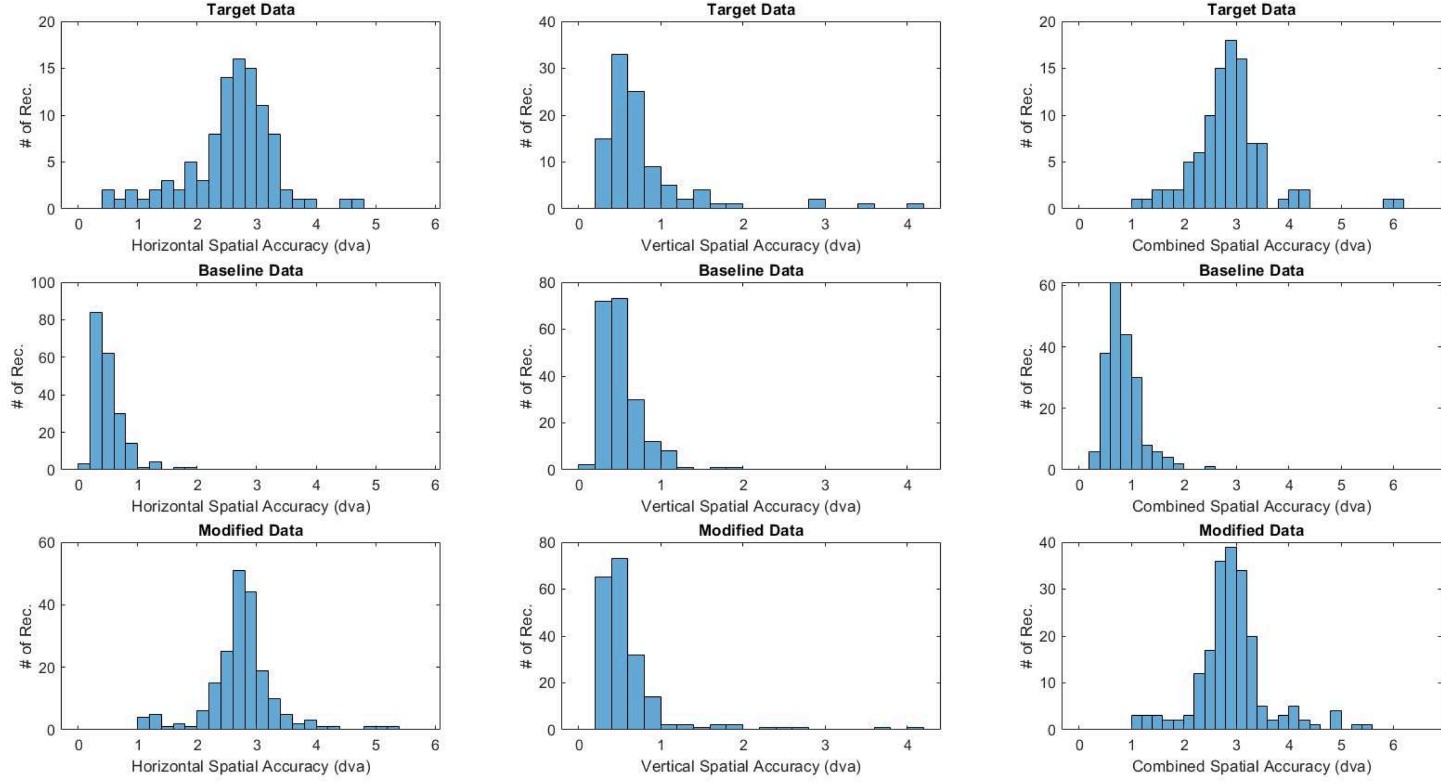

**Fig 4**. Variation in spatial accuracy across the target and synthetic data sets produced by the baseline and modified model, respectively.

pin. When the results obtained from the percentile-matching method are used to estimate the target set performance, the mean absolute error was 12.08%, which results in a substantial improvement over the 31.10% achieved with the baseline. As in d-prime's case, the percentile-matching was also better at estimating the target FRR compared to the baseline in every tested scenario.

The picture starts to differ when the EER is considered. The percentile-matching still outperformed the baseline on average, achieving the mean absolute estimation error of 3.08% compared to 4.69% in the baseline's case. However, the percentile-matching was better only for input sequence lengths in the range from 1 to 7, and 11 (i.e., in 8 cases out of 12 total). To investigate this discrepancy, we inspected the genuine versus impostor similarity score distributions for data degraded with percentile-matching. It turned out that the distribution of genuine scores started to form a distinct lower mode at the sequence length of 6, which continuously enlarged towards the length of 12. This phenomenon was significantly more pronounced in the percentile-matching data, unlike the rest of the sets. In other words, the percentile-matching method produced significantly more genuine embedding pairs with low similarity scores, especially for larger inputs. We hypothesize that this happened due to the deficiency of the marginal degradation parameter selection. Namely, the Session 1 recording used for training and the test one from Session 2 would have inevitable data quality difference. As they come from the same subject, their embedding pair is a part of the genuine distribution. The source data represents a high level of signal quality and the absolute differences in spatial precision are typically small there. However, if the corresponding difference in spatial precision is *relatively* big, it gets unrealistically amplified through the percentile-matching specifically with the switch to a worse spatial precision target profile. It is reasonable to assume that a

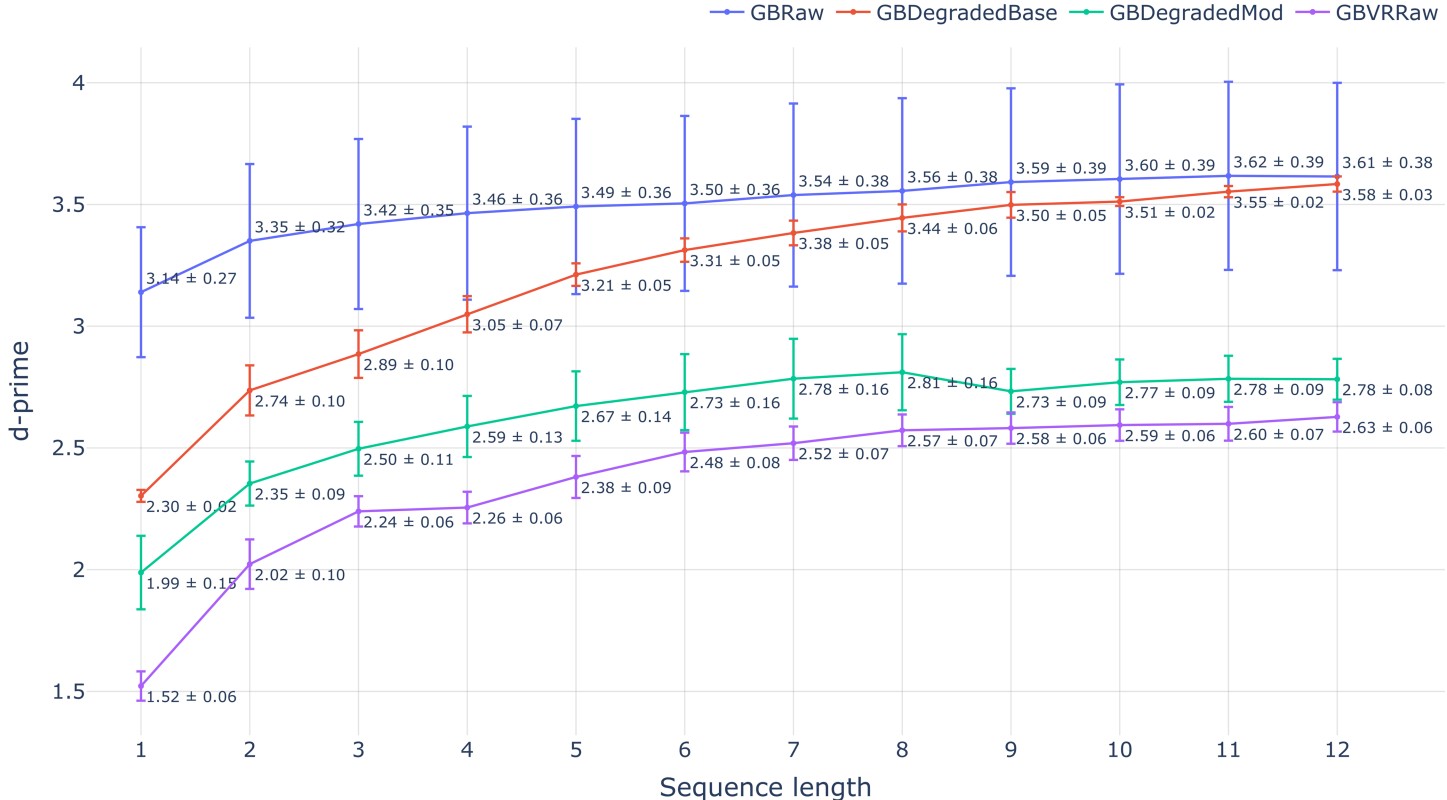

**Fig 5**. Variation in d-prime across different input sequence lengths for the source, target, and synthetic data sets produced by the baseline and modified models.

model might still find the spatial precision to be a good biometric marker for the rest majority of data, and therefore assign a low similarity score in our case.

### Per-recording realism assessment

The 1-NN classification results are summarized in Table 1, where the central tendency and dispersion across the five trials are represented using median and range, respectively. The modified model greatly improves the resulting 1-NN accuracy towards the ideal performance metric of 50%, with the combined (i.e., real and synthetic sample) median classification accuracy reduced by 35.7%. For the modified model, classification accuracy is considerably higher than real sample accuracy, indicating that the synthetic samples tend to be clustered in the ET signal quality vector space.

### Conclusions and future work

The percentile-matching technique for degrading the signal quality of previously captured eye tracking data was proposed and demonstrated herein. The model was shown to produce an improved synthetic approximation of real target data with lower signal quality level. The proposed method produced a distributional structure in the signal quality metrics with significantly enhanced similarity to the target data set compared to the baseline transformation model. This improvement on per-metric level was further validated in a biometric case study chosen to test a real-world practicality of synthetic data degradation. Our method outperformed the baseline in estimating the task-specific performance on real target data with reduced average error in a) d-prime from 0.84 to 0.24; b) FRR (at a 1-in-10000 FAR) from 31.10% to 12.08%; and c) EER

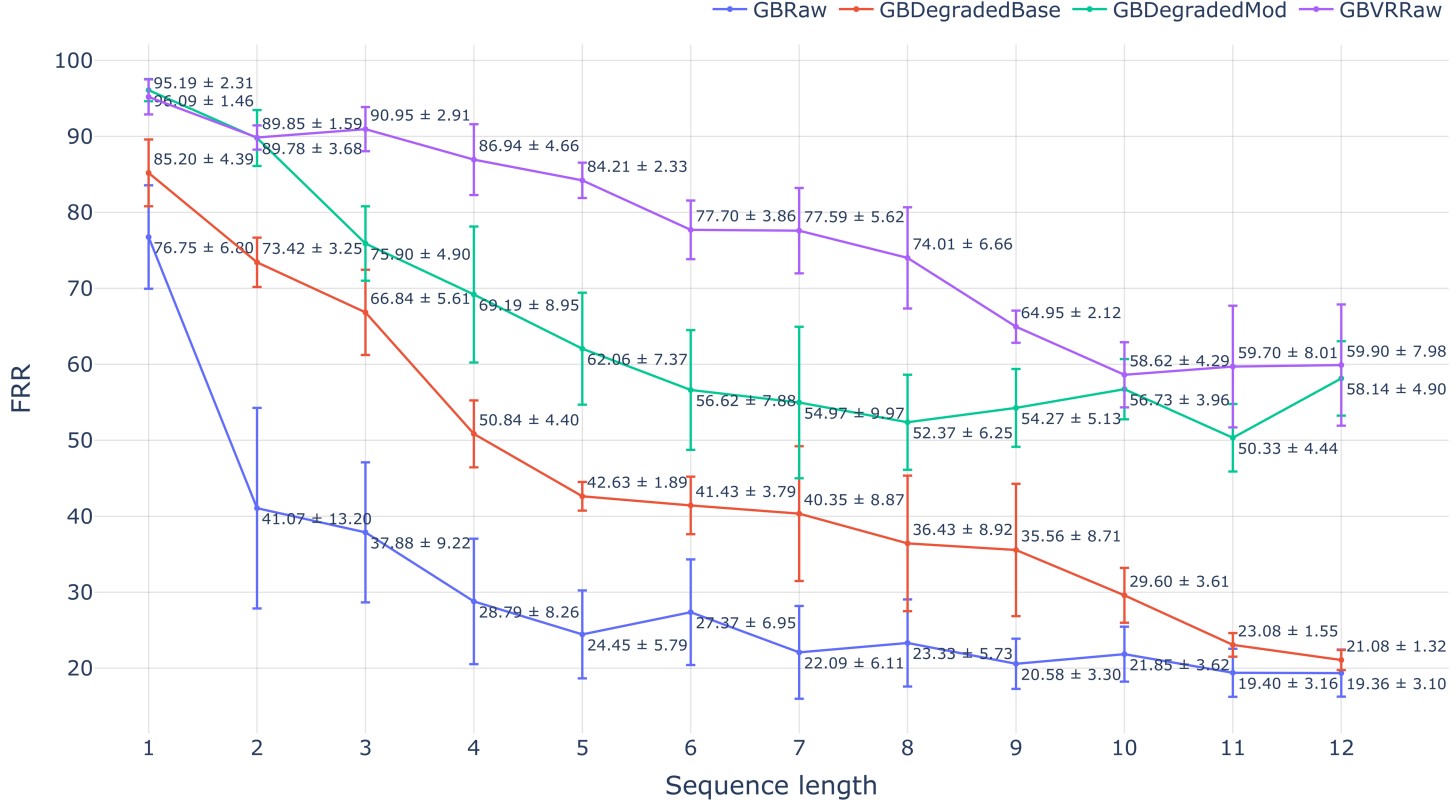

**Fig 6**. Variation in FRR across different input sequence lengths for the source, target, and synthetic data sets produced by the baseline and modified models.

from 4.69% to 3.08%. Only in 4 out of 12 EER evaluation scenarios (out of 36 total), the percentile-matching performed worse. This was attributed to its unrealistic amplification of spatial precision variability between train and test recordings for outlier subjects. More work should be done in the future to investigate the intra-subject across-recording data quality variance and a proper way to incorporate it in a signal degradation model. We hypothesize that the increased similarity between the synthetic and target data will also support data augmentation strategies for other applications such as eye movement classification [44] and gaze prediction [45].

We further expand the literature by generalizing to the signal domain a 1-NN classifier method that was previously applied to image data synthesized with generative adversarial networks. This technique is particularly valuable for the assessment of synthetically generated data for machine learning applications in an application-agnostic framework. While application-based metrics still serve as the ultimate benchmark for a given end task, we hypothesize that the 1-NN algorithm can provide value in rapid prototyping of novel synthetic degradation methods by removing the extra burden of having to retrain the application-specific model on the augmented data set. If found necessary, the richness of the proposed assessment workflow may be further enhanced through the introduction of additional features within the summary feature vector that are not related to considered signal quality metrics.

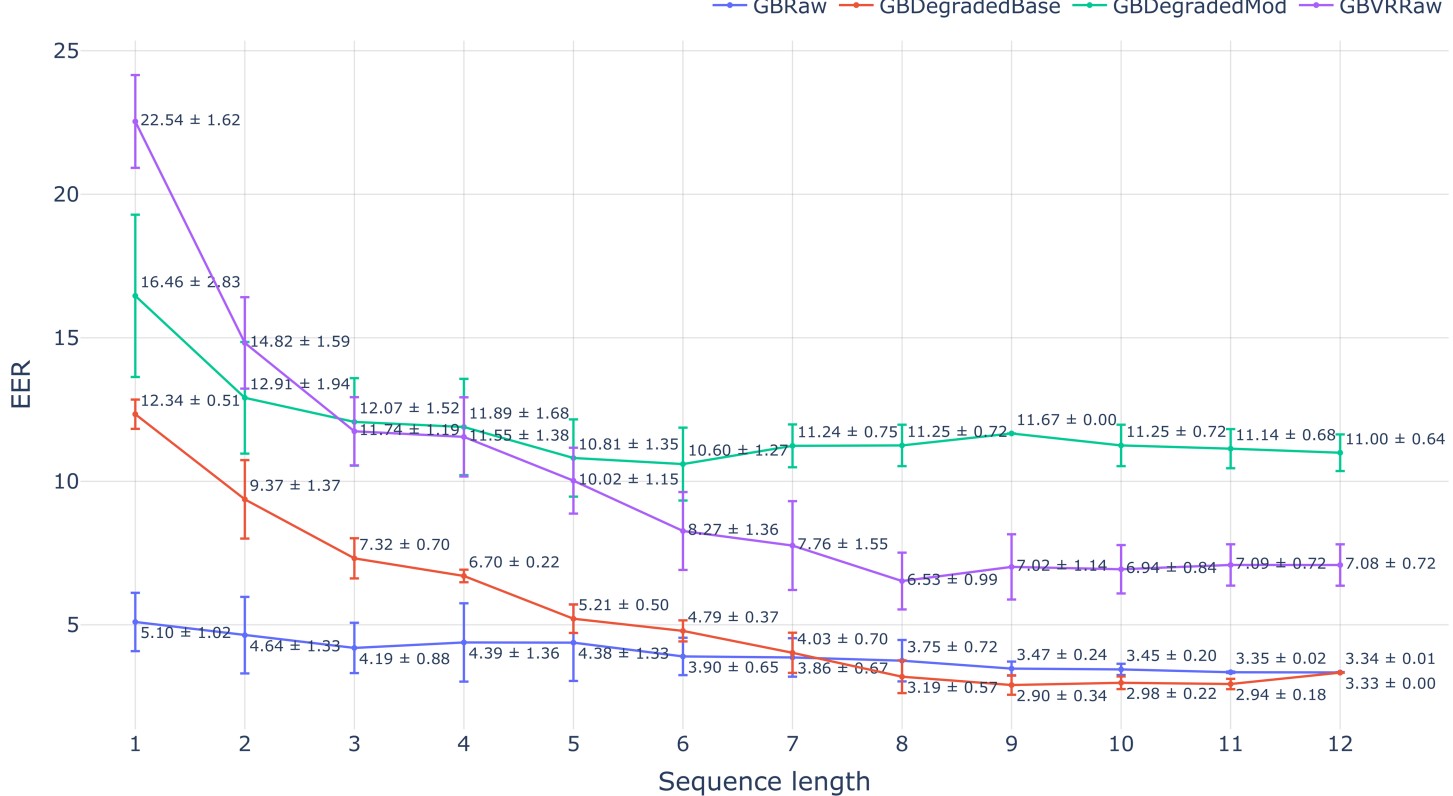

**Fig 7**. Variation in EER across different input sequence lengths for the source, target, and synthetic data sets produced by the baseline and modified models.

**Table 1**. 1-NN classification accuracy results (median $\pm$ range values across 5 random samples; 50% ideal benchmark).

| Metric | Baseline | Modified |
|---|---|---|
| Combined Classification Accuracy | $98.7 \pm 0.5\%$ | $63.0 \pm 7.1\%$ |
| Real Sample Accuracy | $98.9 \pm 0.5\%$ | $54.8 \pm 11.7\%$ |
| Synthetic Sample Accuracy | $98.4 \pm 0.1\%$ | $71.3 \pm 6.4\%$ |

## Author contributions

**Conceptualization:** Henry Griffith.

**Data curation:** Samantha Aziz.

**Funding acquisition:** Oleg V. Komogortsev.

**Investigation:** Henry Griffith, Mehedi H. Raju.

**Methodology:** Henry Griffith, Dmytro Katrychuk, Dillon J. Lohr.

**Project administration:** Oleg V. Komogortsev.

**Resources:** Oleg V. Komogortsev.

**Software:** Henry Griffith, Dillon J. Lohr.

**Validation:** Dmytro Katrychuk, Mehedi H. Raju, Samantha Aziz.

**Visualization:** Henry Griffith, Dmytro Katrychuk.

**Writing – original draft:** Henry Griffith.

**Writing – review & editing:** Dmytro Katrychuk, Samantha Aziz, Dillon J. Lohr.

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
