## [Editor Report · Decision Letter 0]

16 Aug 2024

PONE-D-24-31041The percentile-matching technique for synthetic eye tracking signal degradation: a biometric case studyPLOS ONE

Dear Dr. Katrychuk,

Thank you for submitting your manuscript to PLOS ONE. After careful consideration, we feel that it has merit but does not fully meet PLOS ONE’s publication criteria as it currently stands. Therefore, we invite you to submit a revised version of the manuscript that addresses the points raised during the review process.

This study demonstrates an improved model-based approach for synthetic degradation of pre-recorded eye movement signals. I reckon that the issue of eye tracking and like any other biometric are still contentious about there applications especially in end-user authentication due to varying laws and the tendencies of being lost and used for malicious activities.

The addition to knowledge of this study is strong on the improvement on data degradation using synthetic data for a case study. All the same, there was no information on subjecting this outcome to long-term/simulated integrity test.

This work to the best of my knowledge followed known scientific rigours in carrying out the experiment and result presentation and reporting. What I found largely missing or confused is why the authors used a modified model as benchmark, when it has not ben tested and proven. Also, there is the need to comparatively compare the result with existing literature to provide through empirical evidence the addition to knowledge and contributions to the scientific community. Doing this will go a long way in situating the work in comparison with existing works, thus helping to further the research.

Nevertheless, I commend the authors for their efforts.

We look forward to receiving your revised manuscript.

Kind regards,

Elochukwu Ukwandu, PhD

Academic Editor

PLOS ONE

Journal Requirements:

"Authors Samantha Aziz and Dillon Lohr are recipients of National Science Foundation Graduate Research Fellowship under Grant No. DGE-1840989 and DGE-1144466, respectively."

"Authors Samantha Aziz and Dillon Lohr are recipients of National Science Foundation 

Graduate Research Fellowship under Grant No. DGE-1840989 and DGE-1144466, 

respectively. Any opinion, findings, and conclusions or recommendations expressed in 

this material are those of the authors(s) and do not necessarily reflect the views of the

National Science Foundation. "

"Authors Samantha Aziz and Dillon Lohr are recipients of National Science Foundation Graduate Research Fellowship under Grant No. DGE-1840989 and DGE-1144466, respectively."

Additional Editor Comments:

The percentile-matching technique for synthetic eye tracking signal degradation: a biometric case study

This study demonstrates an improved model-based approach for synthetic degradation of pre-recorded eye movement signals. I reckon that the issue of eye tracking and like any other biometric

are still contentious about there applications especially in end-user authentication due to varying laws and the tendencies of being lost and used for malicious activities.

The addition to knowledge of this study range from the improvement on data degradation using synthetic data for a case study. All the same, there was no information on subjecting this outcome to long-term/simulated integrity test.

This work to the best of my knowledge followed known scientific rigours in carrying out the experiment and result presentation and reporting. What I found largely missing or confused is why the authors used a modified model as benchmark, when it has not ben tested and proven to be trusted. Also, there is the need to comparatively compare the results with existing literature to provide through empirical evidence the addition to knowledge and contributions to the scientific community. Doing this help to situate the work properly for further studies by an interested party.

Nevertheless, I commend the authors on their efforts.

---

## [Author Response · Author response to Decision Letter 1]

20 Feb 2025

This study demonstrates an improved model-based approach for synthetic degradation of pre-recorded eye movement signals. I reckon that the issue of eye tracking and like any other biometric are still contentious about there applications especially in end-user authentication due to varying laws and the tendencies of being lost and used for malicious activities.

> Thank you.

The addition to knowledge of this study range from the improvement on data degradation using synthetic data for a case study. All the same, there was no information on subjecting this outcome to long-term/simulated integrity test. This work to the best of my knowledge followed known scientific rigours in carrying out the experiment and result presentation and reporting.

> Thank you.

What I found largely missing or confused is why the authors used a modified model as benchmark, when it has not been tested and proven to be trusted.

> The mechanistic data degradation model described in the paper titled – “Using machine learning to detect events in eye-tracking data” was implemented herein for initial benchmarking purposes. While it was attempted to replicate the implementation as closely as possible to the procedure described in the original manuscript, some modifications were made to enhance performance for the current application as described in the section- Benchmark transformation model in the manuscript.

Zemblys R, Niehorster DC, Komogortsev O, Holmqvist K. Using machine learning to detect events in eye-tracking data. Behavior research methods. 2018;50(1):160–181.

Also, there is the need to comparatively compare the results with existing literature to provide through empirical evidence the addition to knowledge and contributions to the scientific community. Doing this help to situate the work properly for further studies by an interested party. Nevertheless, I commend the authors on their efforts.

> In table 1 of the manuscript, we compare the results in terms of three metrics. Also In figure 4 we compare the results between baseline and modified versions from the biometric perspective.

---

## [Decision Letter · Decision Letter 1]

21 Mar 2025

PONE-D-24-31041R1The percentile-matching technique for synthetic eye tracking signal degradation: a biometric case studyPLOS ONE

Dear Dr. Katrychuk,

Thank you for submitting your manuscript to PLOS ONE. After careful consideration, we feel that it has merit but does not fully meet PLOS ONE’s publication criteria as it currently stands. Therefore, we invite you to submit a revised version of the manuscript that addresses the points raised during the review process.

Minor Revision required.

We look forward to receiving your revised manuscript.

Kind regards,

Elochukwu Ukwandu, PhD

Academic Editor

PLOS ONE

Journal Requirements:

Reviewers' comments:

Reviewer's Responses to Questions

**Comments to the Author**

1. If the authors have adequately addressed your comments raised in a previous round of review and you feel that this manuscript is now acceptable for publication, you may indicate that here to bypass the “Comments to the Author” section, enter your conflict of interest statement in the “Confidential to Editor” section, and submit your "Accept" recommendation.

Reviewer #1: All comments have been addressed

Reviewer #2: (No Response)

2. Is the manuscript technically sound, and do the data support the conclusions?

Reviewer #1: Yes

Reviewer #2: (No Response)

3. Has the statistical analysis been performed appropriately and rigorously?

Reviewer #1: Yes

Reviewer #2: (No Response)

4. Have the authors made all data underlying the findings in their manuscript fully available?

Reviewer #1: Yes

Reviewer #2: (No Response)

5. Is the manuscript presented in an intelligible fashion and written in standard English?

Reviewer #1: Yes

Reviewer #2: (No Response)

6. Review Comments to the Author

Reviewer #1: Most comments have been addressed, I appreciate the authors effort. I recommend for acceptance to this point.

Reviewer #2: 1. The clarity of the figures in the manuscript is insufficient, so vector figures are recommended.

2. The structure of the manuscript is somewhat confusing. Should the data set be described in the experimental section?

3. Some related works on target tracking tasks should be discussed in this paper: --Self-supervised deep correlation tracking, --Aligned Spatial-Temporal Memory Network for Thermal Infrared Target Tracking, --Active learning for deep tracking…

4. Experiment validations are not convincing. Some additional experiments need to be conducted to make its conclusion stronger.

7. PLOS authors have the option to publish the peer review history of their article (what does this mean?). If published, this will include your full peer review and any attached files.

Reviewer #1: No

Reviewer #2: No

---

## [Author Response · Author response to Decision Letter 2]

17 Jun 2025

Dear Editor,

Please find the response to each of the comments below:

Reviewer #2: 1. The clarity of the figures in the manuscript is insufficient, so vector figures are recommended.

Thank you for pointing this out. We have replaced figures with high-resolution vector (in PDF/SVG format) to enhance clarity and ensure better readability.

2. The structure of the manuscript is somewhat confusing. Should the data set be described in the experimental section?

Thank you for your comments. We would like to clarify that our manuscript does not contain a section explicitly titled "Experimental Section." Currently, the dataset details are provided within the Methods section, which we intended to include both methodological and experimental setup information. If the reviewer recommends separating this into a distinct "Experimental" section for clarity, we are happy to make that structural adjustment. We would appreciate further guidance on how the reviewer would prefer this information to be organized.

3. Some related works on target tracking tasks should be discussed in this paper: --Self-supervised deep correlation tracking, --Aligned Spatial-Temporal Memory Network for Thermal Infrared Target Tracking, --Active learning for deep tracking…

Thank you for these suggestions. While these works represent important advancements in generic object tracking under RGB or thermal modalities, they differ considerably from the focus of our work.

4. Experiment validations are not convincing. Some additional experiments need to be conducted to make its conclusion stronger.

We appreciate the reviewer’s concern regarding the strength of the experimental validation. To better address this point and ensure our revisions align with the reviewer’s expectations, we would greatly value any specific suggestions or directions regarding the types of additional experiments that would most effectively strengthen our conclusions.

---

## [Decision Letter · Decision Letter 2]

30 Sep 2025

`The percentile-matching technique for synthetic eye tracking signal degradation: a biometric case study

PONE-D-24-31041R2

Dear Dr. Katrychuk,

We’re pleased to inform you that your manuscript has been judged scientifically suitable for publication and will be formally accepted for publication once it meets all outstanding technical requirements.

Kind regards,

Elochukwu Ukwandu, PhD

Academic Editor

PLOS ONE

Additional Editor Comments (optional):

Reviewers' comments:

Reviewer's Responses to Questions

**Comments to the Author**

1. If the authors have adequately addressed your comments raised in a previous round of review and you feel that this manuscript is now acceptable for publication, you may indicate that here to bypass the “Comments to the Author” section, enter your conflict of interest statement in the “Confidential to Editor” section, and submit your "Accept" recommendation.

Reviewer #2: (No Response)

Reviewer #3: (No Response)

2. Is the manuscript technically sound, and do the data support the conclusions?

Reviewer #2: (No Response)

Reviewer #3: (No Response)

3. Has the statistical analysis been performed appropriately and rigorously?

Reviewer #2: (No Response)

Reviewer #3: (No Response)

4. Have the authors made all data underlying the findings in their manuscript fully available?

Reviewer #2: (No Response)

Reviewer #3: (No Response)

5. Is the manuscript presented in an intelligible fashion and written in standard English?

Reviewer #2: (No Response)

Reviewer #3: (No Response)

6. Review Comments to the Author

Reviewer #2: The author's research direction is different from mine. I'm not an expert in the field of this paper and I refuse to comment on this manuscript.

Reviewer #3: (No Response)

7. PLOS authors have the option to publish the peer review history of their article (what does this mean?). If published, this will include your full peer review and any attached files.

Reviewer #2: No

Reviewer #3: No

---

## [Editor Report · Acceptance letter]

PONE-D-24-31041R2

PLOS ONE

Dear Dr. Katrychuk,

I'm pleased to inform you that your manuscript has been deemed suitable for publication in PLOS ONE. Congratulations! Your manuscript is now being handed over to our production team.

Kind regards,

on behalf of

Dr. Elochukwu Ukwandu

Academic Editor

PLOS ONE